# Exact Combinatorial Optimization
# with Graph Convolutional Neural Networks

**Maxime Gasse**
Mila, Polytechnique Montréal
`maxime.gasse@polymtl.ca`

**Didier Chételat**
Polytechnique Montréal
`didier.chetelat@polymtl.ca`

**Nicola Ferroni**
University of Bologna
`n.ferroni@specialvideo.it`

**Laurent Charlin**
Mila, HEC Montréal
`laurent.charlin@hec.ca`

**Andrea Lodi**
Mila, Polytechnique Montréal
`andrea.lodi@polymtl.ca`

## Abstract

Combinatorial optimization problems are typically tackled by the branch-and-bound paradigm. We propose a new graph convolutional neural network model for learning branch-and-bound variable selection policies, which leverages the natural variable-constraint bipartite graph representation of mixed-integer linear programs. We train our model via imitation learning from the strong branching expert rule, and demonstrate on a series of hard problems that our approach produces policies that improve upon state-of-the-art machine-learning methods for branching and generalize to instances significantly larger than seen during training. Moreover, we improve for the first time over expert-designed branching rules implemented in a state-of-the-art solver on large problems. Code for reproducing all the experiments can be found at `https://github.com/ds4dm/learn2branch`.

## 1   Introduction

Combinatorial optimization aims to find optimal configurations in discrete spaces where exhaustive enumeration is intractable. It has applications in fields as diverse as electronics, transportation, management, retail, and manufacturing [42], but also in machine learning, such as in structured prediction and maximum a posteriori inference [51; 34; 49]. Such problems can be extremely difficult to solve, and in fact most classical NP-hard computer science problems are examples of combinatorial optimization. Nonetheless, there exists a broad range of exact combinatorial optimization algorithms, which are guaranteed to find an optimal solution despite a worst-case exponential time complexity [52]. An important property of such algorithms is that, when interrupted before termination, they can usually provide an intermediate solution along with an optimality bound, which can be a valuable information in theory and in practice. For example, after one hour of computation, an exact algorithm may give the guarantee that the best solution found so far lies within 2% of the optimum, even without knowing what the actual optimum is. This quality makes exact methods appealing and practical, and as such they constitute the core of modern commercial solvers.

In practice, most combinatorial optimization problems can be formulated as mixed-integer linear programs (MILPs), in which case branch-and-bound (B&B) [35] is the exact method of choice. Branch-and-bound recursively partitions the solution space into a search tree, and computes relaxation

bounds along the way to prune subtrees that provably cannot contain an optimal solution. This iterative process requires sequential decision-making, such as *node selection*: selecting the next node to evaluate, and *variable selection*: selecting the variable by which to partition the node's search space [41]. This decision process traditionally follows a series of hard-coded heuristics, carefully designed by experts to minimize the average solving time on a representative set of MILP instances [21]. However, in many contexts it is common to repeatedly solve similar combinatorial optimization problems, e.g., day-to-day production planning and lot-sizing problems [44], which may significantly differ from the set of instances on which B&B algorithms are typically evaluated. It is then appealing to use statistical learning for tuning B&B algorithms automatically for a desired class of problems. However, this line of work raises two challenges. First, it is not obvious how to encode the state of a MILP B&B decision process [4], especially since both search trees and integer linear programs can have a variable structure and size. Second, it is not clear how to formulate a model architecture that leads to rules which can generalize, at least to similar instances but also ideally to instances larger than seen during training.

In this work we propose to address the above challenges by using graph convolutional neural networks. More precisely, we focus on variable selection, also known as the *branching problem*, which lies at the core of the B&B paradigm yet is still not well theoretically understood [41], and adopt an imitation learning strategy to learn a fast approximation of *strong branching*, a high-quality but expensive branching rule. While such an idea is not new [30; 4; 24], we propose to address the learning problem in a novel way, through two contributions. First, we propose to encode the branching policies into a graph convolutional neural network (GCNN), which allows us to exploit the natural bipartite graph representation of MILP problems, thereby reducing the amount of manual feature engineering. Second, we approximate strong branching decisions by using behavioral cloning with a cross-entropy loss, a less difficult task than predicting strong branching scores [4] or rankings [30; 24]. We evaluate our approach on four classes of NP-hard problems, namely set covering, combinatorial auction, capacitated facility location and maximum independent set. We compare against the previously proposed approaches of Khalil et al. [30], Alvarez et al. [4] and Hansknecht et al. [24], as well as against the default hybrid branching rule in SCIP [20], a modern open-source solver. The results show that our choice of model, state encoding, and training procedure leads to policies that can offer a substantial improvement over traditional branching rules, and generalize well to larger instances than those used in training.

In Section 2, we review the broader literature of works that use statistical learning for branching. In Section 3, we formally introduce the B&B framework, and formulate the branching problem as a Markov decision process. In Section 4, we present our state representation, model, and training procedure for addressing the branching problem. Finally, we discuss experimental results in Section 5.

## 2 Related work

First steps towards statistical learning of branching rules in B&B were taken by Khalil et al. [30], who learn a branching rule customized to a single instance during the B&B process, as well as Alvarez et al. [4] and Hansknecht et al. [24] who learn a branching rule offline on a collection of similar instances, in a fashion similar to us. In each case a branching policy is learned by imitation of the strong branching expert, although with a differently formulated learning problem. Namely, Khalil et al. [30] and Hansknecht et al. [24] treat it as a ranking problem and learn a partial ordering of the candidates produced by the expert, while Alvarez et al. [4] treat it as a regression problem and learn directly the strong branching scores of the candidates. In contrast, we treat it as a classification problem and simply learn from the expert decisions, which allows imitation from experts that don't rely on branching scores or orderings. These works also differ from ours in three other key aspects. First, they rely on extensive feature engineering, which is reduced by our graph convolutional neural network approach. Second, they do not evaluate generalization ability to instances larger than seen during training, which we propose to do. Finally, in each case performance was evaluated on a simplified solver, whereas we compare, for the first time and favorably, against a full-fledged solver with primal heuristics, cuts and presolving activated. We compare against these approaches in Section 5.

Other works have considered using graph convolutional neural networks in the context of approximate combinatorial optimization, where the objective is to find good solutions quickly, without seeking any optimality guarantees. The first work of this nature was by Khalil et al. [31], who proposed a GCNN model for learning greedy heuristics on several collections of combinatorial optimization

problems defined on graphs. This was followed by Selsam et al. [47], who proposed a recurrent GCNN model, NeuroSAT, which can be interpreted as an approximate SAT solver when trained to predict satisfiability. Such works provide additional evidence that GCNNs can effectively capture structural characteristics of combinatorial optimization problems.

Other works consider using machine learning to improve variable selection in branch-and-bound, without directly learning a branching policy. Di Liberto et al. [15] learn a clustering-based classifier to pick a variable selection rule at every branching decision up to a certain depth, while Balcan et al. [8] use the fact that many variable selection rules in B&B explicitly score the candidate variables, and propose to learn a weighting of different existing scores to combine their strengths. Other works learn variable selection policies, but for algorithms less general than B&B. Liang et al. [39] learn a variable selection policy for SAT solvers using a bandit approach, and Lederman et al. [36] extend their work by taking a reinforcement learning approach with graph convolutional neural networks. Unlike our approach, these works are restricted to conflict-driven clause learning methods in SAT solvers, and cannot be readily extended to B&B methods for arbitrary mixed-integer linear programs. In the same vein, Balunovic et al. [9] learn by imitation learning a variable selection procedure for SMT solvers that exploits specific aspects of this type of solver.

Finally, researchers have also focused on learning other aspects of B&B algorithms than variable selection. He et al. [25] learn a node selection heuristic by imitation learning of the oracle procedure that expands the node whose feasible set contains the optimal solution, while Song et al. [48] learn node selection and pruning heuristics by imitation learning of shortest paths to good feasible solutions, and Khalil et al. [32] learn primal heuristics for B&B algorithms. Those approaches are complementary with our work, and could in principle be combined to further improve solver performance. More generally, many authors have proposed machine learning approaches to fine-tune exact optimization algorithms, not necessarily for MILPs in general. A recent survey is provided by Bengio et al. [10].

## 3 Background

### 3.1 Problem definition

A mixed-integer linear program is an optimization problem of the form

$$\arg \min_{\mathbf{x}} \left\{ \mathbf{c}^\top \mathbf{x} \mid \mathbf{A}\mathbf{x} \leq \mathbf{b}, \ \mathbf{l} \leq \mathbf{x} \leq \mathbf{u}, \ \mathbf{x} \in \mathbb{Z}^p \times \mathbb{R}^{n-p} \right\}, \tag{1}$$

where $\mathbf{c} \in \mathbb{R}^n$ is called the objective coefficient vector, $\mathbf{A} \in \mathbb{R}^{m \times n}$ the constraint coefficient matrix, $\mathbf{b} \in \mathbb{R}^m$ the constraint right-hand-side vector, $\mathbf{l}, \mathbf{u} \in \mathbb{R}^n$ respectively the lower and upper variable bound vectors, and $p \leq n$ the number of integer variables. Under this representation, the size of a MILP is typically measured by the number of rows ($m$) and columns ($n$) of the constraint matrix. By relaxing the integrality constraint, one obtains a continuous linear program (LP) whose solution provides a lower bound to (1), and can be solved efficiently using, for example, the simplex algorithm. If a solution to the LP relaxation respects the original integrality constraint, then it is also a solution to (1). If not, then one may decompose the LP relaxation into two sub-problems, by splitting the feasible region according to a variable that does not respect integrality in the current LP solution $\mathbf{x}^\star$,

$$x_i \leq \lfloor x_i^\star \rfloor \lor x_i \geq \lceil x_i^\star \rceil, \quad \exists i \leq p \mid x_i^\star \notin \mathbb{Z}, \tag{2}$$

where $\lfloor . \rfloor$ and $\lceil . \rceil$ respectively denote the floor and ceil functions. In practice, the two sub-problems will only differ from the parent LP in the variable bounds for $x_i$, which get updated to $u_i = \lfloor x_i^\star \rfloor$ in the left child and $l_i = \lceil x_i^\star \rceil$ in the right child.

The branch-and-bound algorithm [52, Ch. II.4], in its simplest formulation, repeatedly performs this binary decomposition, giving rise to a search tree. By design, the best LP solution in the leaf nodes of the tree provides a lower bound to the original MILP, whereas the best integral LP solution (if any) provides an upper bound. The solving process stops whenever both the upper and lower bounds are equal or when the feasible regions do not decompose anymore, thereby providing a certificate of optimality or infeasibility, respectively.

### 3.2 Branching rules

A key step in the B&B algorithm is selecting a fractional variable to branch on in (2), which can have a very significant impact on the size of the resulting search tree [2]. As such, branching rules are at

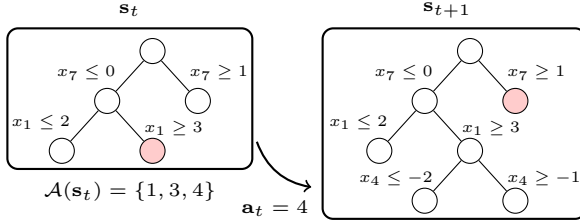

Figure 1: B&B variable selection as a Markov decision process. On the left, a state $\mathbf{s}_t$ comprised of the branch-and-bound tree, with a leaf node chosen by the solver to be expanded next (in pink). On the right, a new state $\mathbf{s}_{t+1}$ resulting from branching on the variable $\mathbf{a}_t = x_4$.

the core of modern combinatorial optimization solvers, and have been the focus of extensive research [40; 43; 1; 17]. So far, the branching strategy consistently resulting in the smallest B&B trees is *strong branching* [5]. It does so by computing the expected bound improvement for each candidate variable before branching, which unfortunately requires the solution of two LPs for every candidate. In practice, running strong branching at every node is prohibitive, and modern B&B solvers instead rely on *hybrid branching* [3; 1] which computes strong branching scores only at the beginning of the solving process and gradually switches to simpler heuristics such as: the conflict score (in the original article), the pseudo-cost [43] or a hand-crafted combination of the two. For a more extensive discussion of B&B branching strategies in MILP, the reader is referred to Achterberg et al. [3].

## 3.3 Markov decision process formulation

As remarked by He et al. [25], the sequential decisions made during B&B can be assimilated to a Markov decision process [26]. Consider the solver to be the environment, and the brancher the agent. At the $t^{\text{th}}$ decision the solver is in a state $\mathbf{s}_t$, which comprises the B&B tree with all past branching decisions, the best integer solution found so far, the LP solution of each node, the currently focused leaf node, as well as any other solver statistics (such as, for example, the number of times every primal heuristic has been called). The brancher then selects a variable $\mathbf{a}_t$ among all fractional variables $\mathcal{A}(\mathbf{s}_t) \subseteq \{1, \ldots, p\}$ at the currently focused node, according to a policy $\pi(\mathbf{a}_t \,|\, \mathbf{s}_t)$. The solver in turn extends the B&B tree, solves the two child LP relaxations, runs any internal heuristic, prunes the tree if warranted, and finally selects the next leaf node to split. We are then in a new state $\mathbf{s}_{t+1}$, and the brancher is called again to take the next branching decision. This process, illustrated in Figure 1, continues until the instance is solved, i.e., until there are no leaf node left for branching.

As a Markov decision process, B&B is episodic, where each episode amounts to solving a MILP instance. Initial states correspond to an instance being sampled among a group of interest, while final states mark the end of the optimization process. The probability of a trajectory $\tau = (\mathbf{s}_0, \ldots, \mathbf{s}_T) \in \mathcal{T}$ then depends on both the branching policy $\pi$ and the remaining components of the solver,

$$p_\pi(\tau) = p(\mathbf{s}_0) \prod_{t=0}^{T-1} \sum_{\mathbf{a} \in \mathcal{A}(\mathbf{s}_t)} \pi(\mathbf{a} \,|\, \mathbf{s}_t) p(\mathbf{s}_{t+1} \,|\, \mathbf{s}_t, \mathbf{a}).$$

A natural approach to find good branching policies is reinforcement learning, with a carefully designed reward function. However, this raises several key issues which we circumvent by adopting an imitation learning scheme, as discussed next.

## 4 Methodology

We now describe our approach for tackling the B&B variable selection problem in MILPs, where we use imitation learning and a dedicated graph convolutional neural network model. As the B&B variable selection problem can be formulated as a Markov decision process, a natural way of training a policy would be reinforcement learning [50]. However, this approach runs into many issues. Notably, as episode length is proportional to performance, and randomly initialized policies perform poorly, standard reinforcement learning algorithms are usually so slow early in training as to make total training time prohibitively long. Moreover, once the initial state corresponding to an instance is selected, the rest of the process is instance-specific, and so the Markov decision processes tend to be extremely large. In this work we choose instead to learn directly from an expert branching rule, an approach usually referred to as imitation learning [27].

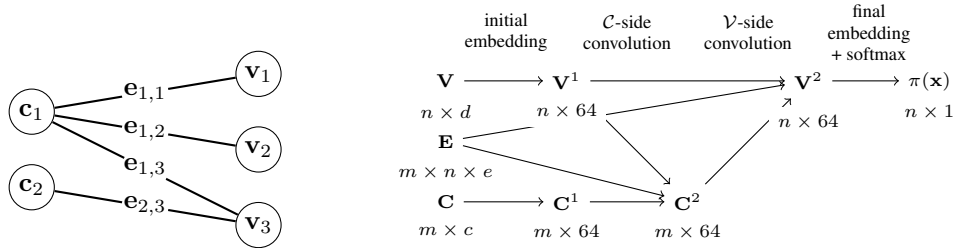

Figure 2: Left: our bipartite state representation $\mathbf{s}_t = (\mathcal{G}, \mathbf{C}, \mathbf{E}, \mathbf{V})$ with $n = 3$ variables and $m = 2$ constraints. Right: our bipartite GCNN architecture for parametrizing our policy $\pi_\theta(\mathbf{a} \mid \mathbf{s}_t)$.

## 4.1 Imitation learning

We train by behavioral cloning [45] using the strong branching rule, which suffers a high computational cost but usually produces the smallest B&B trees, as mentioned in Section 3.2. We first run the expert on a collection of training instances of interest, record a dataset of expert state-action pairs $\mathcal{D} = \{(\mathbf{s}_i, \mathbf{a}_i^\star)\}_{i=1}^N$, and then learn our policy by minimizing the cross-entropy loss

$$\mathcal{L}(\theta) = -\frac{1}{N} \sum_{(\mathbf{s}, \mathbf{a}^*) \in \mathcal{D}} \log \pi_\theta(\mathbf{a}^* \mid \mathbf{s}). \tag{3}$$

## 4.2 State encoding

We encode the state $\mathbf{s}_t$ of the B&B process at time $t$ as a bipartite graph with node and edge features $(\mathcal{G}, \mathbf{C}, \mathbf{E}, \mathbf{V})$, described in Figure 2 (Left). On one side of the graph are nodes corresponding to the constraints in the MILP, one per row in the current node's LP relaxation, with $\mathbf{C} \in \mathbb{R}^{m \times c}$ their feature matrix. On the other side are nodes corresponding to the variables in the MILP, one per LP column, with $\mathbf{V} \in \mathbb{R}^{n \times d}$ their feature matrix. An edge $(i, j) \in \mathcal{E}$ connects a constraint node $i$ and a variable node $j$ if the latter is involved in the former, that is if $\mathbf{A}_{ij} \neq 0$, and $\mathbf{E} \in \mathbb{R}^{m \times n \times e}$ represents the (sparse) tensor of edge features. Note that under mere restrictions in the B&B solver (namely, by enabling cuts only at the root node), the graph structure is the same for all LPs in the B&B tree, which reduces the cost of feature extraction. The exact features attached to the graph are described in the supplementary materials. We note that this is really only a subset of the solver state, which technically turns the process into a partially-observable Markov decision process [6], but also that excellent variable selection policies such as strong branching are able to do well despite relying only on a subset of the solver state as well.

## 4.3 Policy parametrization

We parametrize our variable selection policy $\pi_\theta(\mathbf{a}|\mathbf{s}_t)$ as a graph convolutional neural network [23; 46; 12]. Such models, also known as message-passing neural networks [19], are extensions of convolutional neural networks from grid-structured data (as in images or sounds) to arbitrary graphs. They have been successfully applied to a variety of machine learning tasks with graph-structured inputs, such as prediction of molecular properties [16; 19], program verification [38], and document classification in citation networks [33]. Graph convolutions exhibit many properties which make them a natural choice for graph-structured data in general, and MILP problems in particular: 1) they are well-defined no matter the input graph size; 2) their computational complexity is directly related to the density of the graph, which makes it an ideal choice for processing typically sparse MILP problems; and 3) they are permutation-invariant, that is they will always produce the same output no matter the order in which the nodes are presented.

Our model takes as input our bipartite state representation $\mathbf{s}_t = (\mathcal{G}, \mathbf{C}, \mathbf{V}, \mathbf{E})$ and performs a single graph convolution, in the form of two interleaved half-convolutions. In detail, because of the bipartite structure of the input graph, our graph convolution can be broken down into two successive passes,

one from variable to constraints and one from constraints to variables. These passes take the form

$$\mathbf{c}_i \leftarrow \mathbf{f}_{\mathcal{C}}\Big(\mathbf{c}_i, \sum_j^{(i,j)\in\mathcal{E}} \mathbf{g}_{\mathcal{C}}\left(\mathbf{c}_i, \mathbf{v}_j, \mathbf{e}_{i,j}\right)\Big), \qquad \mathbf{v}_j \leftarrow \mathbf{f}_{\mathcal{V}}\Big(\mathbf{v}_j, \sum_i^{(i,j)\in\mathcal{E}} \mathbf{g}_{\mathcal{V}}\left(\mathbf{c}_i, \mathbf{v}_j, \mathbf{e}_{i,j}\right)\Big) \qquad (4)$$

for all $i \in \mathcal{C}$, $j \in \mathcal{V}$, where $\mathbf{f}_{\mathcal{C}}$, $\mathbf{f}_{\mathcal{V}}$, $\mathbf{g}_{\mathcal{C}}$ and $\mathbf{g}_{\mathcal{V}}$ are 2-layer perceptrons with *relu* activation functions. Following this graph-convolution layer, we obtain a bipartite graph with the same topology as the input, but with potentially different node features, so that each node now contains information from its neighbors. We obtain our policy by discarding the constraint nodes and applying a final 2-layer perceptron on variable nodes, combined with a masked softmax activation to produce a probability distribution over the candidate branching variables (i.e., the non-fixed LP variables). The right side of Figure 2 provides an overview of our architecture.

**Prenorm layers**    In the literature of GCNN, it is common to normalize each convolution operation by the number of neighbours [33]. As noted by Xu et al. [53] this might result in a loss of expressiveness, as the model then becomes unable to perform a simple counting operation (e.g., in how many constraints does a variable appears). Therefore we opt for un-normalized convolutions. However, this introduces a weight initialization issue. Indeed, weight initialization in standard CNNs relies on the number of input units to normalize the initial weights [22], which in a GCNN is unknown beforehand and depends on the dataset. To overcome this issue and stabilize the learning procedure, we adopt a simple affine transformation $\mathbf{x} \leftarrow (\mathbf{x} - \beta)/\sigma$, which we call a *prenorm* layer, applied right after the summation in (4). The $\beta$ and $\sigma$ parameters are initialized with respectively the empirical mean and standard deviation of $\mathbf{x}$ on the training dataset, and fixed once and for all before the actual training. Adopting both un-normalized convolutions and this pre-training procedure improves our generalization performance on larger problems, as will be shown in Section 5.

## 5   Experiments

We now present a comparative experiment against three competing machine learning approaches and SCIP's default branching rule to assess the value of our approach, as well as an ablation study to validate our architectural choices. Code for reproducing all the experiments can be found at `https://github.com/ds4dm/learn2branch`.

### 5.1   Setup

**Benchmarks**    We evaluate our approach on four NP-hard problem benchmarks. Our first benchmark is comprised of set covering instances generated following Balas and Ho [7], with 1,000 columns. We train and test on instances with 500 rows, and we evaluate on instances with 500 (Easy), 1,000 (Medium) and 2,000 (Hard) rows. Our second benchmark is comprised of combinatorial auction instances, generated following the *arbitrary relationships* procedure of Leyton-Brown et al. [37, Section 4.3]. We train and test on instances with 100 items for 500 bids, and we evaluate on instances with 100 items for 500 bids (Easy), 200 items for 1,000 bids (Medium) and 300 items for 1,500 bids (Hard). Our third benchmark is comprised of capacitated facility location instances generated following Cornuejols et al. [14], with 100 facilities. We train and test on instances with 100 customers, and we evaluate on instances with 100 (Easy), 200 (Medium) and 400 (Hard) customers. Finally, our fourth benchmark is comprised of maximum independent set instances on Erdős-Rényi random graphs, generated following the procedure of Bergman et al. [11, 4.6.4] with affinity set to 4. We train and test on instances with graphs of 500 nodes, and we evaluate on instances with 500 (Easy), 1000 (Medium) and 1500 nodes (Hard). These four benchmarks were chosen because they are challenging for state-of-the-art solvers, but also representative of the types of integer programming problems encountered in practice. In particular, set covering problems capture the quintessence of integer linear programming, since column generation formulations can be written for virtually any difficult discrete optimization problem. Throughout all experiments we use SCIP 6.0.1 as the backend solver, with a time limit of 1 hour. Following Karzan et al. [29], Fischetti and Monaci [17] and Khalil et al. [30] we allow cutting plane generation at the root node only, and deactivate solver restarts. All other SCIP parameters are kept to default so as to make comparisons as fair and reproducible as possible.

**Baselines**    We compare against a human-designed state-of-the-art branching rule: reliability pseudocost (RPB), a variant of hybrid branching [1] which is used by default in SCIP. For completeness,

Table 1: Imitation learning accuracy on the test sets.

| | Set Covering | | | Combinatorial Auction | | | Capacitated Facility Location | | | Maximum Independent Set | | |
|---|---|---|---|---|---|---|---|---|---|---|---|---|
| model | acc@1 | acc@5 | acc@10 | acc@1 | acc@5 | acc@10 | acc@1 | acc@5 | acc@10 | acc@1 | acc@5 | acc@10 |
| TREES | 51.8±0.3 | 80.5±0.1 | 91.4±0.2 | 52.9±0.3 | 84.3±0.1 | 94.1±0.1 | 63.0±0.4 | 97.3±0.1 | 99.9±0.0 | 30.9±0.4 | 47.4±0.3 | 54.6±0.3 |
| SVMRANK | 57.6±0.2 | 84.7±0.1 | 94.0±0.1 | 57.2±0.2 | 86.9±0.2 | 95.4±0.1 | 67.8±0.1 | 98.1±0.0 | 99.9±0.0 | 48.0±0.6 | 69.3±0.2 | 78.1±0.2 |
| LMART | 57.4±0.2 | 84.5±0.1 | 93.8±0.1 | 57.3±0.3 | 86.9±0.2 | 95.3±0.1 | 68.0±0.2 | 98.0±0.0 | 99.9±0.0 | 48.9±0.3 | 68.9±0.4 | 77.0±0.5 |
| GCNN | **65.5**±0.1 | **92.4**±0.1 | **98.2**±0.0 | **61.6**±0.1 | **91.0**±0.1 | **97.8**±0.1 | **71.2**±0.2 | **98.6**±0.1 | 99.9±0.0 | **56.5**±0.2 | **80.8**±0.3 | **89.0**±0.1 |

Table 2: Policy evaluation on separate instances in terms of solving time, number of wins (fastest method) over number of solved instances, and number of resulting B&B nodes (lower is better). For each problem, the models are trained on easy instances only. See Section 5.1 for definitions.

| | Easy | | | Medium | | | Hard | | |
|---|---|---|---|---|---|---|---|---|---|
| Model | Time | Wins | Nodes | Time | Wins | Nodes | Time | Wins | Nodes |
| FSB | 17.30 ± 6.1% | 0/100 | 17 ±13.7% | 411.34 ± 4.3% | 0/90 | 171 ± 6.4% | 3600.00 ± 0.0% | 0/0 | n/a ± n/a % |
| RPB | 8.98 ± 4.8% | 0/100 | **54** ±20.8% | 60.07 ± 3.7% | 0/100 | 1741 ± 7.9% | 1677.02 ± 3.0% | 4/65 | 47 299 ± 4.9% |
| TREES | 9.28 ± 4.9% | 0/100 | 187 ± 9.4% | 92.47 ± 5.9% | 0/100 | 2187 ± 7.9% | 2869.21 ± 3.2% | 0/35 | 59 013 ± 9.3% |
| SVMRANK | 8.10 ± 3.8% | 1/100 | 165 ± 8.2% | 73.58 ± 3.1% | 0/100 | 1915 ± 3.8% | 2389.92 ± 2.3% | 0/47 | 42 120 ± 5.4% |
| LMART | 7.19 ± 4.2% | 14/100 | 167 ± 9.0% | 59.98 ± 3.9% | 0/100 | 1925 ± 4.9% | 2165.96 ± 2.0% | 0/54 | 45 319 ± 3.4% |
| GCNN | **6.59** ± 3.1% | **85**/100 | 134 ± 7.6% | **42.48** ± 2.7% | **100**/100 | **1450** ± 3.3% | **1489.91** ± 3.3% | **66**/70 | **29 981** ± 4.9% |

Set Covering

| | Easy | | | Medium | | | Hard | | |
|---|---|---|---|---|---|---|---|---|---|
| Model | Time | Wins | Nodes | Time | Wins | Nodes | Time | Wins | Nodes |
| FSB | 4.11 ±12.1% | 0/100 | 6 ±30.3% | 86.90 ± 12.9% | 0/100 | 72 ±19.4% | 1813.33 ± 5.1% | 0/68 | 400 ± 7.5% |
| RPB | 2.74 ± 7.8% | 0/100 | **10** ±32.1% | 17.41 ± 6.6% | 0/100 | 689 ±21.2% | 136.17 ± 7.9% | 13/100 | 5511 ±11.7% |
| TREES | 2.47 ± 7.3% | 0/100 | 86 ±15.9% | 23.70 ± 11.2% | 0/100 | 976 ±14.4% | 451.39 ±14.6% | 0/95 | 10 290 ±16.2% |
| SVMRANK | 2.31 ± 6.8% | 0/100 | 77 ±15.0% | 23.10 ± 9.8% | 0/100 | 867 ±13.4% | 364.48 ± 7.7% | 0/98 | 6329 ± 7.7% |
| LMART | **1.79** ± 6.0% | **75**/100 | 77 ±14.9% | 14.42 ± 9.5% | 1/100 | 873 ±14.3% | 222.54 ± 8.6% | 0/100 | 7006 ± 6.9% |
| GCNN | 1.85 ± 5.0% | 25/100 | 70 ±12.0% | **10.29** ± 7.1% | **99**/100 | **657** ±12.2% | **114.16** ±10.3% | **87**/100 | **5169** ±14.9% |

Combinatorial Auction

| | Easy | | | Medium | | | Hard | | |
|---|---|---|---|---|---|---|---|---|---|
| Model | Time | Wins | Nodes | Time | Wins | Nodes | Time | Wins | Nodes |
| FSB | 30.36 ±19.6% | 4/100 | 14 ±34.5% | 214.25 ± 15.2% | 1/100 | 76 ±15.8% | 742.91 ± 9.1% | 15/90 | 55 ± 7.2% |
| RPB | 26.55 ±16.2% | 9/100 | **22** ±31.9% | 156.12 ± 11.5% | 8/100 | **142** ±20.6% | 631.50 ± 8.1% | 14/96 | **110** ±15.5% |
| TREES | 28.96 ±14.7% | 3/100 | 135 ±20.0% | 159.86 ± 15.3% | 3/100 | 401 ±11.6% | 671.01 ±11.1% | 1/95 | 381 ±11.1% |
| SVMRANK | 23.58 ±14.1% | 11/100 | 117 ±20.5% | 130.86 ± 13.6% | 13/100 | 348 ±11.4% | 586.13 ±10.0% | 21/95 | 321 ± 8.8% |
| LMART | 23.34 ±13.6% | 16/100 | 117 ±20.7% | 128.48 ± 15.4% | 23/100 | 349 ±12.9% | 582.38 ±10.5% | 15/95 | 314 ± 7.0% |
| GCNN | **22.10** ±15.8% | **57**/100 | 107 ±21.4% | **120.94** ± 14.2% | **52**/100 | 339 ±11.8% | **563.36** ±10.7% | **30**/95 | 338 ±10.9% |

Capacitated Facility Location

| | Easy | | | Medium | | | Hard | | |
|---|---|---|---|---|---|---|---|---|---|
| Model | Time | Wins | Nodes | Time | Wins | Nodes | Time | Wins | Nodes |
| FSB | 23.58 ±29.9% | 9/100 | 7 ±35.9% | 1503.55 ± 20.9% | 0/74 | 38 ±28.2% | 3600.00 ± 0.0% | 0/0 | n/a ± n/a % |
| RPB | 8.77 ±11.8% | 7/100 | **20** ±36.1% | **110.99** ± 24.4% | 41/100 | **729** ±37.3% | 2045.61 ±18.3% | 22/42 | **2675** ±24.0% |
| TREES | 10.75 ±22.1% | 1/100 | 76 ±44.2% | 1183.37 ± 34.2% | 1/47 | 4664 ±45.8% | 3565.12 ± 1.2% | 0/3 | 38 296 ± 4.1% |
| SVMRANK | 8.83 ±14.9% | 2/100 | 46 ±32.2% | 242.91 ± 29.3% | 1/96 | 546 ±26.0% | 2902.94 ± 9.6% | 1/18 | 6256 ±15.1% |
| LMART | 7.31 ±12.7% | 30/100 | 52 ±38.1% | 219.22 ± 36.0% | 15/91 | 747 ±35.1% | 3044.94 ± 7.0% | 0/12 | 8893 ± 3.5% |
| GCNN | **6.43** ±11.6% | **51**/100 | 43 ±40.2% | 192.91 ±110.2% | **42**/82 | 1841 ±88.0% | **2024.37** ±30.6% | **25**/29 | 2997 ±26.3% |

Maximum Independent Set

we report as well the performance of full strong branching (FSB), our slow expert. We also compare against three machine learning branchers: the learning-to-score approach of Alvarez et al. [4] (TREES) based on an ExtraTrees [18] model, as well as the learning-to-rank approaches from Khalil et al. [30] (SVMRANK) and Hansknecht et al. [24] (LMART), based on an SVMrank [28] and a LambdaMART [13] model, respectively. The TREES model uses variable-wise features from our bipartite state, obtained by concatenating the variable's node features with edge and constraint node features statistics over its neighborhood. The SVMRANK and LMART models both use the original features proposed by Khalil et al. [30], which we re-implemented within SCIP. More training details for each machine learning method can be found in the supplementary materials.

**Training** We train the models on each benchmark separately. Namely, for each benchmark, we generate 100,000 branching samples extracted from 10,000 randomly generated instances for training, 20,000 branching samples from 2,000 instances for validation, and same for test (see supplementary materials for details). We report in Table 1 the test accuracy of each machine learning model over five seeds, as the percentage of times the highest ranked decision of the model (acc@1), one of the five highest (acc@5) and one of the ten highest (acc@10) is a variable which is given the highest strong branching score.

**Evaluation**   Evaluation is performed for each problem difficulty (Easy, Medium, Hard) on 20 new instances using five different seeds[1], which amounts to a total of 100 solving attempts per method. We report standard metrics for MILP benchmarking[2], that is: the 1-shifted geometric mean of the solving times in seconds, including running times of unsolved instances without extra penalization (Time); the hardware-independent final node counts of instances that are solved by all baselines (Nodes); and the number of times each branching policy results in the fastest solving time, over the number of instances solved (Win). Policy evaluation results are displayed in Table 2. Note that we also report the average per-instance standard deviation, so "$64 \pm 13.6\%$ nodes" means it took on average 64 nodes to solve an instance, and when solving one of those instances the number of nodes varied by 13.6% on average.

## 5.2   Comparative experiment

In terms of prediction accuracy (Table 1), GCNN clearly outperforms the baseline competitors on all four problems, while SVMRANK and LMART are on par with each other and the performance of TREES is the lowest.

At solving time (Table 2), the accuracy of each method is clearly reflected in the number of nodes required to solve the instances. Interestingly, the best method in terms of nodes is not necessarily the best in terms of total solving time, which also takes into account the computational cost of each branching policy, i.e., the feature extraction and inference time. The SVMRANK approach, despite being slightly better than LMART in terms of number of nodes, is also slower due to a worse running time / number of nodes trade-off. Our GCNN model clearly dominates overall, except on combinatorial auction (Easy) and maximum independent set (Medium) instances, where LMART and RPB are respectively faster.

Our GCNN model generalizes well to instances of size larger than seen during training, and outperforms SCIP's default branching rule RPB in terms of running time in almost every configuration. In particular and strikingly, it significantly outperforms RPB in terms of nodes on medium and hard instances for setcover and combinatorial auction problems. As expected, the FSB expert brancher is not competitive in terms of running time, despite producing very small search trees. The maximum independent set problem seems particularly challenging for generalization, as all machine learning approaches report a lower number of solved instances than the default RPB brancher, and GCNN, despite being the fastest machine learning approach overall, exhibits a high variability both in terms of time and number of nodes.

For the first time in the literature a machine-learning-based approach is compared with an essentially full-fledged MILP solver. For this reason, the results are particularly impressive, and indicate that GCNN is a very serious candidate to be implemented within a MILP solver, as an additional tool to speed up mixed-integer linear programming solvers. Also, they suggest that more could be gained from a tight integration within a complex software, like any MILP solver is.

## 5.3   Ablation study

We present an ablation study of our proposed GCNN model on the set covering problem by comparing three variants of the convolution operation in (4): mean rather than sum convolutions (MEAN), sum convolutions without our prenorm layer (SUM) and finally sum convolutions with prenorm layers, which is the model we use throughout our experiments (GCNN).

Results on test instances are reported in Table 3. The solving performance of both variants MEAN and SUM is very similar to that of our baseline GCNN on small instances. On large instances however, the variants perform significantly worse in terms of both solving time and number of nodes, especially on hard instances. This empirical evidence supports our hypothesis that sum-convolutions offer a better architectural prior than mean-convolution from the task of learning to branch, and that our prenorm layer helps for stabilizing training and improving generalization.

Table 3: Ablation study of our GCNN model on the set covering problem. Sum convolutions generalize better to larger instances, especially when combined with a prenorm layer.

| Model | Accuracies | | | Easy | | | Medium | | | Hard | | |
|---|---|---|---|---|---|---|---|---|---|---|---|---|
| | acc@1 | acc@5 | acc@10 | time | wins | nodes | time | wins | nodes | time | wins | nodes |
| MEAN | 65.4 ±0.1 | **92.4** ±0.1 | **98.2** ±0.0 | 6.7 ±3% | 13 / 100 | 134 ±6% | 43.7 ±3% | 19 / 100 | 1894 ±4% | 1593.0 ±4% | 6 / 70 | 62 227 ±6% |
| SUM | 65.5 ±0.2 | 92.3 ±0.2 | 98.1 ±0.1 | **6.6** ±3% | 27 / 100 | 134 ±6% | **42.5** ±3% | **45** / 100 | 1882 ±4% | 1511.7 ±3% | 22 / 70 | 57 864 ±4% |
| GCNN | **65.5** ±0.1 | **92.4** ±0.1 | **98.2** ±0.0 | **6.6** ±3% | **60** / 100 | 134 ±8% | **42.5** ±3% | 36 / 100 | **1870** ±3% | **1489.9** ±3% | **42** / 70 | **56 348** ±5% |

# 6  Discussion

The objective of branch-and-bound is to solve combinatorial optimization problems as fast as possible. Branching policies must therefore balance the quality of decisions taken with the time spent to take each decision. An extreme example of this tradeoff is strong branching: this policy takes excellent decisions leading to low number of nodes overall, but every decision-making step is so slow that the overall running time is not competitive. Early experiments showed that we could take better decisions and decrease the number of nodes slightly on average by training a GCNN policy with more layers or with a larger embedding size. However, this would also lead to increased computational costs for inference and slightly larger times at each decision, and in the end increased solving times on average. The policy architecture we propose is thus a compromise between learning capacity and inference speed, something that is not traditionally a concern within the machine learning community.

Another concern among the combinatorial optimization community is the ability of policies trained on small instances to generalize to larger instances. We were able to show that machine learning methods, and the GCNN model in particular, can generalize to fairly larger instances. However, in general it is expected that the improvement in performance decreases as our model is evaluated on progressively larger problems, as can already be observed from Table 2. In early experiments with even larger instances (huge), we observed a performance drop for the model trained on our small instances. This could presumably be remedied by training on larger instances in the first place, and indeed a model trained on medium instances did perform well on those huge instances again. In any case, there are limits as to the generalization ability of any learned branching policy, and since the limit is likely very dependent on the problem structure, it is difficult to give any precise quantitative estimates a priori. This desirable ability to generalize outside of the training distribution, sometimes termed transfer learning, is also not a traditional concern in the machine learning community.

# 7  Conclusion

We formulated branch-and-bound, the standard exact method for solving mixed-integer linear programs, as a Markov decision process. In this context, we proposed and evaluated a novel approach for tackling the branching problem, by expressing the state of the branch-and-bound process as a bipartite graph, which reduces the need for feature engineering by naturally leveraging the variable-constraint structure of MILP problems, and allows for the encoding of branching policies as a graph convolutional neural network. We demonstrated on four NP-hard problems that, by adopting a simple imitation learning scheme, the policies learned by a GCNN were outperforming previously proposed machine learning approaches for branching, and could also outperform the default branching strategy of SCIP, a modern open-source solver. Most importantly, we demonstrated that the learned policies could generalize to instance sizes larger than seen during training. This is essential since collecting strong branching decisions, hence training, can be computationally prohibitive on large instances. Our work indicates that the GCNN model, especially using sum convolutions with the proposed prenorm layer, is a good architectural prior for the task of branching in MILP.

In future work, we would like to assess the viability of our approach on a broader set on combinatorial problems, and also experiment with reinforcement learning methods for improving over the policies learned by imitation. Also, we believe that there is plenty of room for hybrid approaches combining traditional methods and machine learning for branching, and we would like to dig deeper into the learned policies in order to extract some knowledge of interest for the MILP community.

## Acknowledgements

We would like to thank the anonymous reviewers whose contributions helped considerably improve the quality of this paper. We would also like to thank Ambros Gleixner and Benjamin Müller for enlightening discussions and technical help regarding SCIP, as well as Gonzalo Muñoz, Aleksandr Kazachkov and Giulia Zarpellon for insightful discussions on variable selection. Finally, we thank Jason Jo, Meng Qu and Mike Pieper for their helpful comments on the structure of the paper.

This work was supported by the Canada First Research Excellence Fund (CFREF), IVADO, CIFAR, GERAD, and Canada Excellence Research Chairs (CERC).

## Footnotes

[1]In addition to ML models which are re-trained with a different seed, all major MILP solvers have a parameter, *seed*, that randomizes some tie-breaking rules, so as to be able to report aggregated results over the same instance.

[2]See e.g. `http://plato.asu.edu/bench.html`

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
