[Supplementary Material]

# Exact Combinatorial Optimization
# with Graph Convolutional Neural Networks
# Supplementary Materials

**Maxime Gasse**
Mila, Polytechnique Montréal
maxime.gasse@polymtl.ca

**Didier Chételat**
Polytechnique Montréal
didier.chetelat@polymtl.ca

**Nicola Ferroni**
University of Bologna
n.ferroni@specialvideo.it

**Laurent Charlin**
Mila, HEC Montréal
laurent.charlin@hec.ca

**Andrea Lodi**
Mila, Polytechnique Montréal
andrea.lodi@polymtl.ca

## 1 Dataset collection details

For each benchmark problem, namely, set covering, combinatorial auction, capacitated facility location and maximum independent set, we generate 10,000 random instances for training, 2,000 for validation, and 3x20 for testing (20 easy instances, 20 medium instances, and 20 hard instances). In order to obtain our datasets of state-action pairs $\{(\mathbf{s}_i, \mathbf{a}_i)\}$ for training and validation, we pick an instance from the corresponding set (training or validation), solve it with SCIP, each time with a new random seed, and record new node states and strong branching decision during the branch-and-bound process. We continue processing new instances by sampling with replacement, until the desired number of node samples is reached, that is, 100,000 samples for training, and 20,000 for validation. Note that this way the whole set of training and validation instances is not necessarily used to generate samples. We report the number of training instances actually used for each problem in Table 1.

Table 1: Number of training instances solved by SCIP for obtaining 100,000 training samples (state-action pairs). We report both the total number of SCIP solves and the number of unique instances solved, since instances are sampled with replacement.

|  | Set Covering | Combinatorial Auction | Capacitated Facility Location | Maximum Independent Set |
|---|---|---|---|---|
| total | 7771 | 11 322 | 7159 | 6198 |
| unique | 5335 | 6661 | 5046 | 4349 |

Note that the strong branching rule, as implemented in the SCIP solver, does not only provides branching decisions, but also triggers side-effects which change the state of the solver itself. In order to use strong branching as an oracle only, when generating our training samples, we re-implemented a vanilla version of the full strong branching rule in SCIP, named *vanillafullstrong*. This version of full strong branching also facilitates the extraction of strong branching scores for training the ranking-based and regression-based machine learning competitors, and will be included by default in the next version of SCIP.

## 2   Training details

### 2.1   GCNN

As described in the main paper, for our GCNN model we record strong branching decisions ($\mathbf{a}$) and extract bipartite state representations ($\mathbf{s}$) during branch-and-bound on a collection of training instances. This yields a training dataset of state-action pairs $\{(\mathbf{s}_t, \mathbf{a}_t)\}$. We train our model with the Tensorflow [1] library, with the same procedure throughout all experiments. We first pretrain the prenorm layers as described in Section 4.3 of the main paper. We then minimize a cross-entropy loss using Adam [4] with minibatches of size 32 and an initial learning rate of 1e-3. We divide the learning rate by 5 when the validation loss does not improve for 10 epochs, and stop training if it does not improve for 20. All experiments are performed on a machine with an Intel Xeon Gold 6126 CPU at 2.60GHz and an Nvidia Tesla V100 GPU.

A list of the features included in our bipartite state representation is given as Table 2. For more details the reader is referred to the source code at `https://github.com/ds4dm/learn2branch`.

Table 2: Description of the constraint, edge and variable features in our bipartite state representation $\mathbf{s}_t = (\mathcal{G}, \mathbf{C}, \mathbf{E}, \mathbf{V})$.

| Tensor | Feature | Description |
|---|---|---|
| **C** | obj_cos_sim | Cosine similarity with objective. |
| | bias | Bias value, normalized with constraint coefficients. |
| | is_tight | Tightness indicator in LP solution. |
| | dualsol_val | Dual solution value, normalized. |
| | age | LP age, normalized with total number of LPs. |
| **E** | coef | Constraint coefficient, normalized per constraint. |
| **V** | type | Type (binary, integer, impl. integer, continuous) as a one-hot encoding. |
| | coef | Objective coefficient, normalized. |
| | has_lb | Lower bound indicator. |
| | has_ub | Upper bound indicator. |
| | sol_is_at_lb | Solution value equals lower bound. |
| | sol_is_at_ub | Solution value equals upper bound. |
| | sol_frac | Solution value fractionality. |
| | basis_status | Simplex basis status (lower, basic, upper, zero) as a one-hot encoding. |
| | reduced_cost | Reduced cost, normalized. |
| | age | LP age, normalized. |
| | sol_val | Solution value. |
| | inc_val | Value in incumbent. |
| | avg_inc_val | Average value in incumbents. |

### 2.2   SVMrank and LambdaMART

For SVMrank and LambdaMART, at each node $t$ we record strong branching ranks $\rho_i$ for each candidate variable $i$, and extract the same variable-wise features $\phi_i^{\text{Khalil}}$ as Khalil et al. [3], with two modifications. First, because of differences between CPLEX and SCIP, it is difficult to reimplement single/double infeasibility statistics, but SCIP keeps track of left/right infeasibility statistics, namely the number of times branching on the variable led to a left (resp. right) infeasible child. Because these statistics capture similar aspects of the branch-and-bound tree, we swapped one for the other. Second, since in our setting at test time there is no separate data collection stage, statistics are computed on

past branching decisions at each time step $t$. Otherwise we follow [3], which suggest a zero-one feature normalization over the candidate, a.k.a. *query-based normalization*, and a binarization of the ranking labels around the 80th centile. This yields a training dataset of variable-wise state-rank pairs $\{(\phi_i^{\text{Khalil}}(\mathbf{s}_t), \rho_{i,t})\}$.

The SVMrank model is trained by minimizing a cost sensitive pairwise loss with a 2nd-order polynomial feature augmentation, and regularization on the validation set among $C \in \{0.001, 0.01, 0.1, 1\}$, as in the original implementation. The LambdaMART model uses 500 estimators, and is trained by maximizing a normalized discounted cumulative gain with default parameters, with early stopping using the validation set.

Note that in their SVMrank implementation, Khalil et al. [3] limit at each node the candidate variables to the ten with highest pseudocost, both for training and testing. This makes sense in their online setting, as the model is trained and applied after observing the behavior of strong branching on a few initial nodes, which initializes the pseudocosts properly. In our context, however, there is no initial strong branching phase, and as is to be expected we observed that applying this pseudocost filtering scheme degraded performance. Consequently, we do not filter, and rather train and test using the whole set of candidate variables at every node.

In addition, both the SVMrank and LambdaMART approaches suffer from poorer training scalability than stochastic gradient descent-based deep neural networks, and we found that training on our entire dataset was prohibitive. Indeed, training would have taken more than 500GB of RAM for SVMrank and more than 1 week for LambdaMART for even the smallest problem class. Consequently, we had to limit the size of the dataset, and chose to reduce the training set to 250,000 candidate variables, and the validation set to 100,000.

## 2.3 ExtraTrees

For ExtraTrees, we record strong branching scores $\sigma_i$ for each candidate variable $i$, and extract variable-wise features $\phi_i^{\text{simple}}$ from our bipartite state $\mathbf{s}$ as follows. For every variable we keep the original variable node features $\mathbf{v}_i$, which we concatenate with the component-wise minimum, mean and maximum of the edge and constraint node features $(\mathbf{e}_{i,j}, \mathbf{c}_j)$ over its neighborhood. As a result, we obtain a training dataset of variable-wise state-score pairs $\{(\phi_i^{\text{simple}}(\mathbf{s}_t), \sigma_{i,t})\}$. The model is trained by mean-squared error minimization, while at test time branching is made on the variable with the highest predicted score, $i^* = \arg\max_i \hat{\sigma}(\phi_i(\mathbf{s}_t))$. As Alvarez et al. [2, p. 14] mention, with this model the inference time increases with the training set size. Additionally, even though in practice the ExtraTrees model training scales to larger datasets than the SVMrank and LambdaMART models, here also we ran into memory issues when training on our entire dataset. Consequently, for ExtraTrees we also chose to limit the training dataset to 250,000 candidate variables, a figure roughly in line with Alvarez et al. [2]. Note that we did not use the expert features proposed by Khalil et al. [3] for ExtraTrees, as those were resulting in degraded performance.

## 3 When are decisions hard for the GCNN policy?

To understand what kinds of decisions are taken by the trained GCNN policy beyond the imitation learning accuracy, it is insightful to look at when it is confident. In particular, we can take the entropy of our policy as a measure of confidence, and compute an analogous quantity from the strong branching scores, $\mathbb{H}(\text{strong branching}) = \log m$ for $m$ the number of candidates with maximal strong branching score. Figure 1 shows that entropies at every decision point on the easy instances for the four problems roughly correlate. This suggests that the decisions where the GCNN hesitates are those that are intrinsically difficult, since on those the expert hesitates more as well.

## 4 Training set sizes for the machine learning methods

As discussed, computational limitations made training of machine learning competitors impossible on the full dataset that we used to train our GCNN policy. It is a limitation of those methods that they cannot benefit from as much training data as the proposed deep policy with the same computational budget, and conversely, the scalability of our proposed method to large training sets is a major advantage. Since there is little reason for artificially restricting any model to smaller datasets than

Figure 1: Entropy of the GCNN policy against the "entropy" of strong branching.

Table 3: Number of branching nodes (state-action pairs) used for training the machine learning competitors, that is, for obtaining 250,000 training and 100,000 validation samples (variable-score pairs). In comparison, the number of branching nodes in the complete dataset used to train the GCNN model is 100,000 for training, and 20,000 for validation.

|  | Set Covering | Combinatorial Auction | Capacitated Facility Location | Maximum Independent Set |
|---|---|---|---|---|
| training | 1979 | 1738 | 6201 | 534 |
| validation | 782 | 696 | 2465 | 213 |

they can handle, especially since the state-of-the-art branching rule (reliability pseudocost) is not even a machine learning method, we did not do so in the paper.

Nonetheless, a natural question is whether the improvements in performance provided by the GCNN came solely from increased dataset size. To answer this question, we re-trained a GCNN model using the same amount of data as for the competitors, that is, 250,000 candidate variables for training, and 100,000 for validation. The size of the resulting datasets, from the point of view of the number of branching nodes recorded, is reported in Table 3. As can be seen in Table 4, the resulting model (GCNN-SMALL) still clearly outperforms the competitors in terms of accuracy. Thus although using more training samples improves performance, the improvements cannot be explained only by the amount of training data.

Table 4: Imitation learning accuracy on the test sets.

| | Set Covering | | | Combinatorial Auction | | | Capacitated Facility Location | | | Maximum Independent Set | | |
|---|---|---|---|---|---|---|---|---|---|---|---|---|
| model | acc@1 | acc@5 | acc@10 | acc@1 | acc@5 | acc@10 | acc@1 | acc@5 | acc@10 | acc@1 | acc@5 | acc@10 |
| TREES | 51.8±0.3 | 80.5±0.1 | 91.4±0.2 | 52.9±0.3 | 84.3±0.1 | 94.1±0.1 | 63.0±0.4 | 97.3±0.1 | 99.9±0.0 | 30.9±0.4 | 47.4±0.3 | 54.6±0.3 |
| SVMRANK | 57.6±0.2 | 84.7±0.1 | 94.0±0.1 | 57.2±0.2 | 86.9±0.2 | 95.4±0.1 | 67.8±0.1 | 98.1±0.1 | 99.9±0.0 | 48.0±0.6 | 69.3±0.2 | 78.1±0.2 |
| LMART | 57.4±0.2 | 84.5±0.1 | 93.8±0.1 | **57.3**±0.3 | 86.9±0.2 | 95.3±0.1 | 68.0±0.2 | 98.0±0.0 | 99.9±0.0 | 48.9±0.3 | 68.9±0.4 | 77.0±0.5 |
| GCNN-SMALL | **57.9**±1.0 | **87.1**±0.6 | **95.5**±0.3 | 55.0±1.6 | **88.0**±0.6 | **96.2**±0.1 | **69.1**±0.1 | **98.2**±0.0 | 99.9±0.0 | **50.1**±1.2 | **73.4**±0.6 | **82.8**±0.6 |
| GCNN | 65.5±0.1 | 92.4±0.1 | 98.2±0.0 | 61.6±0.1 | 91.0±0.1 | 97.8±0.1 | 71.2±0.2 | 98.6±0.1 | 99.9±0.0 | 56.5±0.2 | 80.8±0.3 | 89.0±0.1 |

Table 5: Training time for each machine learning method, in hours.

| | Set Covering | Combinatorial Auction | Capacitated Facility Location | Maximum Independent Set |
|---|---|---|---|---|
| TREES | $0.05 \pm 0.00$ | $0.03 \pm 0.00$ | $0.16 \pm 0.01$ | $0.04 \pm 0.00$ |
| SVMRANK | $1.21 \pm 0.01$ | $1.17 \pm 0.06$ | $1.04 \pm 0.03$ | $1.19 \pm 0.02$ |
| LMART | $2.87 \pm 0.23$ | $2.47 \pm 0.26$ | $1.38 \pm 0.15$ | $2.16 \pm 0.53$ |
| GCNN | $14.45 \pm 1.56$ | $3.84 \pm 0.33$ | $18.18 \pm 2.98$ | $4.73 \pm 0.85$ |

# 5 Training and inference times

For completeness, we report in Table 5 the training time of each machine learning method, and in Table 6 their inference time per node. As can be observed, the GCNN model relies on less complicated features than the other methods, and therefore requires less computational time during feature extraction, at the cost of a higher prediction time.

Table 6: Inference time per node for each machine learning model, in milliseconds. We report both the time required to extract and compute features from SCIP (feat. extract), and the total inference time (total) which includes feature extraction and model prediction.

| | Easy | | Medium | | Hard | |
|---|---|---|---|---|---|---|
| model | total | feat. extract | total | feat. extract | total | feat. extract |
| TREES | 23.8 ± 4.0 | 11.2 ± 2.0 | 28.4 ± 4.5 | 15.7 ± 3.2 | 54.9 ±14.2 | 42.7 ±13.8 |
| SVMRANK | 16.6 ± 7.1 | 6.4 ± 4.3 | 23.1 ±11.1 | 9.4 ± 5.1 | 150.4 ±19.3 | 55.6 ±12.6 |
| LMART | 7.0 ± 3.5 | 6.0 ± 3.3 | 10.1 ± 4.8 | 8.8 ± 4.6 | 55.7 ±12.9 | 51.9 ±12.6 |
| GCNN | 5.5 ±13.1 | 1.1 ± 3.5 | 5.4 ± 4.6 | 1.5 ± 3.0 | 10.2 ±12.7 | 4.9 ±10.2 |

Set Covering

| | | | | | | |
|---|---|---|---|---|---|---|
| TREES | 16.1 ± 5.1 | 5.9 ± 1.8 | 25.3 ± 6.1 | 10.6 ± 4.7 | 30.2 ± 9.6 | 15.6 ± 7.8 |
| SVMRANK | 14.7 ±10.4 | 3.9 ± 4.8 | 31.6 ±20.2 | 8.3 ± 5.7 | 55.4 ±39.2 | 13.5 ± 9.4 |
| LMART | 4.5 ± 2.6 | 3.6 ± 2.3 | 9.6 ± 5.7 | 8.2 ± 5.4 | 15.4 ± 9.1 | 13.5 ± 8.6 |
| GCNN | 7.2 ±24.3 | 1.2 ± 5.7 | 4.4 ± 6.1 | 1.5 ± 4.2 | 5.1 ± 6.3 | 2.1 ± 5.3 |

Combinatorial Auction

| | | | | | | |
|---|---|---|---|---|---|---|
| TREES | 53.7 ± 2.0 | 49.1 ± 1.1 | 99.0 ± 6.2 | 93.4 ± 4.6 | 205.3 ± 4.9 | 199.7 ± 4.3 |
| SVMRANK | 13.3 ± 6.5 | 9.3 ± 5.8 | 21.8 ±14.5 | 17.9 ±14.0 | 80.4 ±70.9 | 74.1 ±70.5 |
| LMART | 9.7 ± 6.2 | 9.3 ± 6.1 | 18.4 ±14.1 | 17.9 ±13.9 | 83.5 ±80.4 | 82.9 ±80.3 |
| GCNN | 9.2 ±15.3 | 3.9 ± 3.3 | 14.8 ±18.7 | 7.5 ± 1.1 | 43.0 ±53.7 | 19.9 ±13.9 |

Capacitated Facility Location

| | | | | | | |
|---|---|---|---|---|---|---|
| TREES | 41.3 ± 7.8 | 12.0 ± 5.8 | 56.9 ±12.4 | 25.0 ±11.1 | 83.8 ±17.5 | 45.4 ±15.5 |
| SVMRANK | 45.5 ± 5.6 | 6.9 ± 5.4 | 96.5 ±12.8 | 19.4 ±11.8 | 200.2 ±17.8 | 35.4 ±17.5 |
| LMART | 8.3 ± 5.0 | 6.6 ± 5.0 | 22.1 ±11.2 | 19.2 ±11.2 | 38.8 ±16.0 | 34.8 ±15.9 |
| GCNN | 5.1 ± 7.4 | 2.3 ± 5.1 | 7.7 ±11.1 | 4.8 ±10.0 | 12.0 ±17.5 | 8.1 ±14.8 |

Maximum Independent Set