[Reviews · NeurIPS 2019]

Reviewer 1



Although the paper addresses an important problem and can be of practical use, I have some concerns about the novelty of this submission. Imitation learning (or supervised learning) is the standard techniques used in many applications. Lots of works have used GCNN for different combinatorial optimization problems and achieves good performance, and I didn't see strong novelty here. Furthermore, with limited ablation analysis, we don't have a good understanding on what's going on. With pure imitation learning, did the proposed approach actually generalize better, compared to the original strong branching it is mimicking? Or is the improvement just because of the speedup from the learned network? Another question is that since the trained network can mimic strong branching heuristics, it might have learned something interesting that takes the place of two LP evaluations. I am curious about what it learns. Did it learn easy cases and give random answers to corner cases? Does it actually learn to address hard cases (if so then we could even replace LP with Neural Network)? Code is available, which is great. I roughly checked the code and it is quite clean and easy to understand. Eqn. 3 is very confusing, since the paper doesn't do reinforcement learning (either from scratch or from pre-trained models). I suggest removing all RL components to make the paper clear. No number in Table. 2, #nodes, Set Covering, Hard case. =============== Update after rebuttal. The rebuttal is generally well-written with additional experiments. It also shows that the performance boost is largely due to an increase of the training data (compared to SVMrank etc). While the algorithmic contribution is not that novel, this can serve as a good baseline for ML-based branching method. In addition, the code is really well-written and available.

Reviewer 2



Summary: This paper proposes a method using graph convolutional neural networks for finding a good branching policy for brach-and-bound algorithm for solving mixed integer linear programs. In particular, the proposed methods makes use of a bipartite graph representation of MILPs and uses imitation learning and a dedicated graph convolutional network. The approach is novel as far as I know of. The paper is generally well written, provides necessary background and discusses related work in a sufficient manner. Pros: - Generally well written paper - Experimental evaluation compares to other approaches (branching policies) - Code provided along the submission Cons: - Experiments of crafted instances, would be (in addition) interesting to see how the approach works when using "real-life" benchmarks Minor comments/questions: - In Table 3 there are missing values (set covering, hard, FSB), wrong bolding (set covering, easy, RSB has less nodes that GCNN) and missing bolding (combinatorial auction, easy, wins) =================== After author response: The author response further clarifies many of the issues raised in the reviews. For the additional results of maximum independent set it is hard to evaluate whether these are interesting or not, since no details are provided on the instances used (hence please provide more details in the upcoming versions). I think that in particular the concerns regarding the fairness of the experimental evaluation are very relevant, and the authors should do their best to address them. The response already provides some answers and clarifications, altough a more thorough experimental evaluation could make the paper stronger.

Reviewer 3



Update following rebuttal: thanks for taking the time to run additional experiments and reporting back! I am generally supportive of the paper and as such have increased my score to 7. I hope the updates about related work will be incorporated if the paper is accepted, as well as additional experiments you found added value. Summary: This paper proposes an imitation learning approach for learning a branching strategy for integer programming. Key to this approach is the use of a graph neural network representation of the integer programs, together with feature engineering. This work differs from other recent learning-to-branch approaches in that the learning task, using imitation, might be simpler than previous ranking or regression formulations, and that the graph neural network can capture structural information of the instance beyond the simple handcrafted features of previous work. The resulting (trained) GCNN models outperform both the standard reliability branching strategy and three ML models from previous work on three different types of problems. Additionally, the GCNN is trained on small instances and tested on larger ones without degradation in performance. Significance: This paper contributes to the recent literature on using ML for combinatorial optimization. This is an interesting space and this paper definitely offers something new. However, as I argue in "Originality", the main building blocks of the proposed method are extensions of 2017-2018 work, albeit in a slightly more general setting. Additionally, the experimental evaluation is missing in some aspects which I discuss in "Quality". Originality: The combination of the REINFORCE-like learning formulation and graph neural network for integer programs is the first of its kind. That being said, very similar ideas have been proposed in the last 2 years: - The first use of imitation for branching appears in [a], which the authors should cite, discuss and compare with or use as needed. - The first use of GNN appears in [b] for graph optimization and [c] for SAT. In particular, the model in [c] is essentially identical to the one presented in this paper. The main difference is that the initial embedding in [c] is a random vector, rather than handcrafted features. The generalization from SAT to MIP is straightforward. Otherwise, the literature review is quite thorough and well-organized. Quality: - "handcrafted variable features": The GCNN also uses such features for initialization. To claim that you do not rely on such features, you should test a purely structural GCNN, initialized with random embedding vectors, as in [c]. This ablation study would allow you to quantify the benefit from using the GCNN. This is not to say that your final model should not be as it currently is (hybridizing features + embeddings is great), but claiming that other methods rely on them while GCNN does not is not completely accurate. - Data collection: Do you get 10 nodes per problem, on average, for the 10k problems, to obtain 100k nodes as training samples? It is not clear how the overall data collection strategy works; [a] is focused on that. - Training with the same amount of data: If I understand correctly, TREES, SVMRANK and LMART are all trained with less data than GCNN, due to memory/running time limitations. It is not clear what the impact of this decision on the final results is. For instance, [29] limits the candidate variables per node to the top 10 based on PC, whereas you don't seem to limit the number of candidate variables. Of course, you should not do so for GCNN, since the graph neural net relies on exchange of information between all variables. However, if you limit SVMRANK to 250k variables, then depending on the number of variables per node, you might be exposing SVMRANK to much less information than GCNN. For example, if you don't cap the number of candidate variables to 10 and instead have 100 variables per node, then SVMRANK sees data from only 2.5k nodes compared to the 100k of GCNN. There are two ways to address this: 1) Use the exact same amount of training data in all methods and report the performance of the corresponding trained models; 2) Fix the computation budget to be the same for all methods at training time, but allow the faster/less memory intensive ones to use up as much training data as they can. I believe you tried to do the latter, but since you use a very powerful GPU for GCNN versus a CPU for the other ML methods, this comparison is very tricky. As such, I recommend that you do 1) for fair comparison across all ML methods. Additionally, [29] should also be trained with only 10 candidate variables per node as in that paper. The intuition is that the ranking model should only focus on the more promising variables. Note that I believe that using massive compute power such as a powerful GPU to train the branching model online is totally fine and encouraged. All that matters is inference time speed. I just think the comparison should be with the same amount of data. - Evaluation: I found the evaluation set of 20 instances x 5 seeds to be extremely small, compared to the 10k training instances. Why not test on 1k instances to strengthen the confidence in the running time and node values of Table 1, especially that they are very close for some of the methods for some of the instance sets? - Difficulty: The Easy/Medium instances solve very fast, in < 3 minutes for all but FSB. Even for Hard instances, the difference in time between any top 2 methods is very small and does not exceed 20 seconds. Since you can train on small problems and test on larger ones, I believe that you should evaluate on problems at least 2x larger than Hard (in addition to Hard). This is important because it will show whether GCNN generalizes to really large problems where smarter branching has a very large impact on total running time (i.e. on the order of minutes or tens of minutes). If not, then is it because embedding at inference time becomes prohibitively slow? Could one train the GCNN on larger instances to ease the generalization to even larger instances? Clarity: Overall, the paper is very well-written and easy to read. Here are some minor clarification questions: - Page 5, 201: "perceptrons relu" -> "perceptrons with relu" - Table 1: How do you handle ties? Assume that strong branching results in the same score for variables x1 and x2, you branch on x1 and record that as the "correct" variable, and one of the methods ranks x2 over x1. Does that method get a point in the acc@k calculation? I ask because branching can lead to infeasible children which makes ties quite possible. - Embedding size: Is the variable embedding vector of size equal to the number of features for V in Table 1 of the appendix, i.e. to 13? If so, have you thought of increasing the embedding size through initially random features to potentially improve the accuracy? - Embedding iterations: Is (5) run only once before the final MLP+Softmax? [b] and [c] perform more embedding iterations, e.g. 3-5, and seem to benefit from that; this could also help your case, since there is still a gap to 100% in the acc@1 of Table 1. - Training time: could you report the average time it takes to train the GCNN on GPU? - Inference time: does the GCNN model run on GPU or CPU at test time? References: [a] Song, Jialin, et al. "Learning to search via retrospective imitation." arXiv preprint arXiv:1804.00846 (2018). [b] Dai, Hanjun, et al. "Learning combinatorial optimization algorithms over graphs." Proceedings of the 31st International Conference on Neural Information Processing Systems. Curran Associates Inc., 2017. [c] Selsam, Daniel, et al. "Learning a SAT Solver from Single-Bit Supervision." ICLR, 2019.

[Author Response · NeurIPS 2019]

We are very thankful to all reviewers for their time and valuable comments. In response, we propose to use the additional
page allowed in the camera-ready version to: **I)** expand our literature review to include other related approaches; **II)**
incorporate more details about our dataset generation procedure, as well as training and inference times; **III)** include
a "Discussion" section to answer the questions raised by each reviewer; **IV)** add an additional problem, *maximum*
*independent set*, to our benchmark; **V)** fix the missing value in Table 2 which is confusing, and should be read as N/A
(not available), since FSB did not solve any instance in that case.

**Answer to reviewer #3: I)** We agree that "lots of works have used GCNN for different combinatorial optimization
problems", but we emphasize that combinatorial optimization can mean many things. We are the first to use GCNNs
(as well as behavioral cloning) for *exact* solving, namely by using them to model branching policies: this is novel and
not trivial to do, since one must also come up with a way to encode branch-and-bound states appropriately. It is also
an important problem to tackle, since exact optimization is ubiquitous in industry. Furthermore, we provide strong,
reproducible empirical evidence that this bipartite GCNN is better suited than previously proposed models for this
important task. **II)** Regarding the policy quality vs inference speed: we agree both aspects are important, and comparing
number of nodes vs total running time in Table 2 already illustrates this trade-off. We propose to clarify this question
explicitly in the text, especially regarding time-consuming methods like strong branching. **III)** Regarding the RL part,
we understand the issue, and we propose to keep the MDP formulation but remove any references to rewards, such as
Eq. 3. **IV)** Regarding interpretability of our learned model, that is a hard question to answer: how to define an easy/hard
branching case? We measured the entropy of both FSB and our learned policy and find that they strongly correlate,
indicating that "obvious" choices for FSB seem to be also "obvious" for our model. We will include this result in the
supplementary materials – we hope this can provide more insights on the behavior of the trained policy, as requested.

**Answer to reviewer #4:** We agree that our benchmark is artificial, and using real-world instances would bring value.
Such datasets could be collected, e.g. similar unit commitment problems are solved every 15 minutes by electrical
power production companies: however, no public one is known to us. At the same time, it is common to evaluate OR
algorithms on artificial problems, and we have used for our benchmark peer-reviewed, published instance generation
algorithms for well-recognized problems, which were designed to model real-world applications. Regarding additional
structure, since our original submission we have run experiments on a fourth problem, maximum independent set, and
here again the GCNN dominates (see below). We intend to include those new results in the final version of the paper.

| Model | Easy | | | Medium | | | Hard | | |
|---|---|---|---|---|---|---|---|---|---|
| | Time | Wins | Nodes | Time | Wins | Nodes | Time | Wins | Nodes |
| FSB | 34.82 ±27.2% | 5 / 100 | 7 ±35.7% | 2434.80 ±15.8% | 0 / 52 | 67 ±37.3% | 3600.00 ± 0.0% | 0 / 0 | n/a ± n/a % |
| RPB | 12.01 ±14.2% | 3 / 100 | **20** ±56.0% | 175.00 ±20.0% | 28 / 100 | 1292 ±29.1% | 2759.82 ±11.3% | 11 / 34 | 8156 ±34.4% |
| TREES | 11.77 ±19.9% | 4 / 100 | 79 ±47.6% | 1691.76 ±37.4% | 0 / 44 | 9441 ±58.1% | 3600.03 ± 0.0% | 0 / 0 | n/a ± n/a % |
| SVM | 9.70 ±13.1% | 9 / 100 | 43 ±33.6% | 434.34 ±37.4% | 0 / 80 | **867** ±36.0% | 3499.30 ± 1.8% | 0 / 4 | 10 256 ±12.2% |
| LMART | 8.36 ±11.3% | 18 / 100 | 48 ±39.6% | 318.38 ±33.0% | 6 / 84 | 1042 ±47.4% | 3493.27 ± 2.5% | 0 / 3 | 15 368 ±48.5% |
| GCNN | **7.81** ± 9.0% | **61** / 100 | 38 ±31.2% | **149.12** ±54.8% | **66** / 93 | 955 ±59.0% | **2281.58** ±29.1% | **28** / 32 | **5070** ±79.5% |

Maximum Independent Set

**Answer to reviewer #5: I)** We agree that we should discuss references [a-c] in our literature review. We propose to
expand this section and better compare our approach to those related works. **II)** Regarding feature engineering and
embeddings, we agree and will clarify this point. **III)** Regarding experimental details, we agree and will add a detailed
description of our dataset generation procedure, the total number of variables/nodes/instances represented in each
dataset, the evaluation metrics and training/inference times of each method. **IV)** Regarding evaluation on even bigger
instances, it seems that GCNN (and other ML methods) fail to generalize at some point, e.g., training on easy (100 items
x 500 bids) combinatorial auction instances does not generalize to huge (400 x 2000) instances, however re-training our
model on medium (200 x 1000) does. We propose to add this discussion in the paper. **V)** Regarding evaluation set size,
we agree 20 instances x 5 seeds is small, but benchmarking already takes more than a hundred hours of computing time.
Unfortunately, we do not currently have the resources to do more. **V)** Regarding pseudocost filtering in [29], thank you
for pointing this out (it was a hard catch from the text). Indeed, by filtering the top-10 variables for training and testing
we were able to train competing models using the whole dataset, but the performance was 1.15-1.5x worse in time, and
2-8x worse in number of nodes. This might be because PC scores require an initialization with SB at the beginning of
the solving process, which is present in [29] due to their hybrid strategy, while in our context we are trying to replace SB
completely. PC scores are then very poor, especially at the root of the tree where branching decisions are most crucial.
Nonetheless, we can include such results in the supplementary materials. **VI)** Regarding the unfair reduced training size
for other methods, that is a good point, and we propose to add a discussion in the paper. However, we believe that the
inability to exploit huge amounts of data is an intrinsic limitation of other methods (e.g. it would require >500GB RAM
for SVMrank, >1 week training for LambdaMART), and is not a good reason for limiting ourselves to the same amount
of data. This being said, we also ran preliminary experiments using small datasets (10x smaller), and our performance
was only slightly worse. Therefore we believe that the superiority of GCNN cannot come from the dataset size alone.

Again, we thank the reviewers for their time, and hope our proposed modifications will answer to their concerns.

[Meta-Review · NeurIPS 2019]

The reviewers have converged to support the paper but there are still some issues with the fairness of the experimental evaluation. I would like to strongly encourage the authors to update the paper according to the suggestions.